# Unchanged but undecided: Reproductive intentions among Ghanaian women following COVID-19 lockdowns in two large metropolises in Ghana

Veena Bhagavathi[1], Deda Ogum[2], Kofi Agyabeng[3], Dorcas Coker-Appiah[4], Fidelia N. A. Ohemeng[5]*

**1** Rutgers Robert Wood Johnson Medical School, New Brunswick, New Jersey, United States of America, **2** Department of Population, Family & Reproductive Health, School of Public Health, University of Ghana, Legon, Accra, Ghana, **3** Department of Biostatistics, School of Global Public Health, New York University, New York, New York, United States of America, **4** Gender Studies and Human Rights Documentation Centre, Accra, Ghana, **5** Department of Sociology, University of Ghana, Legon, Accra, Ghana

* fohemeng@ug.edu.gh

## Abstract

Evidence suggests that disaster situations such as the COVID-19 pandemic may affect reproductive intentions. These effects interact with an individual's social context, including socioeconomic status, cultural norms, family size and structure, to shape reproductive intentions. In this paper, we review the COVID-19 lockdowns' effects on reproductive intentions among Ghanaian women living in Accra and Kumasi. A correlational quantitative research design was adopted for the study while data was obtained through a survey of 532 women of reproductive age. Descriptive statistics, bivariate modeling using Chi-square and Fisher's exact tests and multivariate modeling using Firth's logistic regression model were conducted to understand determinants of changing intentions following the COVID-19 lockdowns. In all, 25 women (4.7%) reported changing their intentions. Those who come from Kumasi were more likely to change their intentions (p < 0.001) as were those who reside within their respective metropolis (p < 0.05). Those who had ever been married or cohabitated with a partner (p < 0.01), who have their childrearing decisions made with input from their partner or other family members (p < 0.05), and who were pregnant or expecting to become pregnant before the lockdowns were enforced (p < 0.001) were also more likely to change their minds. Residence in Kumasi (AOR = 4.21, 95%CI 1.41–12.62) and being pregnant or expecting to become pregnant just before lockdown (AOR = 4.29, 95%CI 1.28–14.32) were the strongest predictors for changing of intentions. Overall, reproductive intentions among those surveyed were largely characterized by ambivalence and inconsistency, with the largest groups of people labeling their state prior to COVID-19 lockdowns as "undecided" and their change in state after lockdown also as "undecided." Future work should include more room

**Data availability statement:** The final dataset and accompanying code are available on the Open Science Framework, DOI https://doi.org/10.17605/OSF.IO/NCWS7.

**Funding:** This study was funded by International Network to End Violence against Women and Girls (INEVAWG).

**Competing interests:** The authors have declared that no competing interests exist.

for ambiguity when characterizing reproductive intentions among Ghanaian women. Incorporating room for ambiguity in characterizing reproductive intentions may improve accuracy in data collection in demographic health research and shift how success is measured in family planning policy. We hope that this shift will promote individualized and person-centered care in the realm of family planning.

## Introduction

The COVID-19 pandemic first reached Ghana in March of 2020 and immediately resulted in a swift lockdown in the Greater Accra and Ashanti regions. In the subsequent months and years, the pandemic shook the economic, social, and demographic fabric of the country in unprecedented ways. Some people found their workload increasing, taking on childcare duties on top of remote jobs. Others lost their income entirely, as outdoor markets were shut down. Relationships were affected by social distancing regulations, which pulled some people to different locations while pushing some who were already cohabitating closer together. Women and girls were uniquely affected by the pandemic. Reports revealed increased violence against women and girls as they spent more time at home. Sexual and reproductive healthcare infrastructure was heavily impacted in the initial phases of establishing essential healthcare [1–6].

In the pandemic's early stages, researchers were curious to understand how COVID-19 and its subsequent restrictions would affect fertility intentions, which can be understood as the desire, plan, or intention to have or to not have children. Fertility intentions are shaped by a variety of factors, including parity, religious beliefs or conservatism, education level, and socioeconomic status. These factors may all be upended in crisis situations.

Findings from research regarding the impact of COVID-19 and associated lockdowns on fertility intentions have varied around the world. A multi-country study that included data from 10,000 women across 30 countries found that around 14.4% reported changing intentions, which was comprised of 10% of women surveyed postponing pregnancy and 4.4% of women surveyed accelerating it [7]. In a scoping review of COVID-19 pandemic's impact on unplanned pregnancy, only one of 15 articles suggested a positive relationship between COVID-19 and child-bearing intentions; the rest either indicated a negative relationship, no change, or unclear changes in intentions [8].

Lessons can also be taken from how populations have reacted in the past to other disaster situations. Studies of reproductive rates following the Spanish Flu and Swine Flu epidemics showed an immediate drop in birth rates followed by an increase in fertility, believed in part to be due to a desire to increase family size after losing children and a shift in expectations on the offspring to provide family security [9].

There are few studies of reproductive intentions in Ghana in general, and to our knowledge none studying reproductive intentions specifically during crisis situations. Sex composition of living children [10], level of education, parity, geographic location,

urban versus rural environment of residence, and "internalization of patriarchal norms" have been shown to affect reproductive intentions of Ghanaian women [11]. Notably, qualitative work has shown that reproductive intentions among Ghanaian women are largely characterized by ambivalence and instability [12].

At the time of this study, it was not clear yet to what extent sexual and reproductive healthcare infrastructure would be impacted by the COVID-19 pandemic, but projections showed that nearly half a million Ghanaian women were at risk of losing access to family planning services [13]. We know now that one third of Ghanaian women [6] reported that the lockdown affected their access to family planning or abortion services in some way, but that systems recovered quickly [2].

This study seeks to understand the COVID-19 lockdowns' impact on reproductive intentions among Ghanaian women. Past work studying reproductive intentions in this population has mostly focused on retrospective data from demographic surveys [11,14] rather than stated intentions and has not considered the impact of a situation like a lockdown on intentions. Understanding factors that affect reproductive intention is crucial in meeting Ghana's high unmet need for family planning [15] and in combating the maternal mortality crisis [16], particularly through a reproductive justice lens of meeting individual women's personal intentions. Further, the effect of disasters and crises on sexual and reproductive health and fertility has been well-characterized in high-income settings [17–19] but less so in low- and middle-income settings. We hope that the results of this study will illuminate the extent to which support is needed for sexual and reproductive healthcare in disaster situations in Ghana and similar settings.

This study sought to investigate whether Ghanaian women changed their reproductive intentions because of COVID-19 lockdown measures and why the lockdown measures may have had an impact. In doing so, this study seeks to address the dearth of literature regarding reproductive intentions among Ghanaian women, as well as the gap in the literature regarding reproductive intentions in Ghanaian women specifically during crisis situations.

## Methods

### Study design

This paper analyzes a subset of data from a larger study that aimed to primarily assess the economic and social impact of COVID-19 response measures on women and girls in Accra and Kumasi. The overarching study followed a convergent parallel mixed-methods design and recruited both economically active and inactive women with the study's primary objective in mind. The current analysis is based on data from the quantitative survey and focuses on reproductive decision-making among participants of reproductive age at the time of the survey.

### Study setting

Ghana is a lower-middle-income country in West Africa. Accra and Kumasi, the largest and most economically active cities in Ghana, were chosen as the sites for the study because they recorded the most COVID-19 cases and were therefore targeted with the strictest lockdown measures. Following the first recorded cases in Ghana in March of 2020, Greater Accra and nearby Kasoa and Greater Kumasi were put under lockdown for three weeks. Large gatherings such as funerals, conferences, and sporting events were banned, and international borders were closed. Outdoor marketplaces were closed, leaving thousands out of work. At the time of data collection in March 2021, the Greater Accra and Ashanti regions (home to Accra and Kumasi) had recorded the highest case counts in the country, reaching 42,312 and 13,092 cases respectively [20]. Further, of the total population of Ghana, over one-third of persons live in the Greater Accra and Ashanti regions, which count Accra and Kumasi as their respective economic centers. The Greater Accra and Ashanti regions represent a range of experiences regarding fertility. For example, the teenage pregnancy rates in Greater Accra and Ashanti are 6% and 23.9% respectively, bracketing the national average of 15.2%. Similarly, rates of reported use of any method of contraception are 32% and 44.4% respectively, bracketing the national average of 36.3%. Sampling from these regions allowed for representation of a large swath of Ghana's overall population.

## Recruitment and participants

A total of 564 women were interviewed for the overarching study. Participants (age>=18years) from various backgrounds were recruited from sub-metropolitan areas, including from a database of female business owners operating within the Accra and Kumasi metropolis, to ensure a representative sample of women to meet the study's primary objective of assessing the overall economic and social impact of COVID-19 lockdown measures on women and girls in Accra and Kumasi. The exclusion criteria were non-residence and not being economically engaged in the study area for at least 6 months prior to the beginning of COVID-19 lockdown and 6 months following the lockdown. For the purposes of this report, analysis was restricted to the 532 women of reproductive age at the time of the survey.

## Study tools

During data collection for the overarching study, information was collected using a structured questionnaire divided into five sections: sociodemographic characteristics, state of women and girls (including relationship dynamics and reproductive health), experiences and exposure to violence and injury, economic impact of COVID-19, and psychological impact of COVID-19. The study tool was first piloted in a community in Accra among women with similar characteristics to the target population. Piloting the survey allowed the study team to refine the questionnaire's wording, flow, and reliability before implementing it with the study population. The questionnaire was uploaded on an android device via the ODK collect app for data collection and then interviewer-administered in-person in accordance with social distancing guidelines. Interviews were conducted in the participants' preferred language, whether English or other local dialect (predominantly Twi or Ga), by a study team member fluent in the preferred language. The data analyzed in this report was taken from the sociodemographic characteristics and reproductive health sub-sections of the survey.

## Measures

**Outcome variables.** The outcome variable in this analysis was *change in fertility intentions related to COVID-19 lockdown*, as measured by a question that directly asked, "Did your reproductive status or plans change since the COVID-19 restrictions were imposed?" For those who responded "yes," we asked what changed. We then asked a series of questions regarding reasons for the decision and people involved in the decision, regardless of whether they reported that their fertility intentions had changed.

**Covariates.** Our primary independent variables related to women's demographic and reproductive health characteristics. Demographic variables included age, ethnicity, city of residence, location of residence with respect to the metropolis, highest level of education completed, religion, employment status, and income relative to the group's median income. Reproductive health characteristics included relationship status, marriage or cohabitation history, gravidity, number of living children, who makes decisions about childbearing, reported number of children desired, contraception use, and reproductive status prior to lockdown. All variables were selected based on local context and on existing literature investigating reproductive decision-making (to do with reproductive intentions, family planning uptake, and antenatal and post-natal care utilization) broadly among Ghanaian women.

## Statistical analysis

The data was downloaded from the host site, cleaned, and analyzed using R. We reported frequency and percentages for categorical variables. Age, income level, and number of living children were continuous variables recoded as categorical variables for analysis purposes. We also conducted hypothesis testing using Chi-square and Fisher's exact tests to test for association between categorical independent variables and *change in fertility intentions related to COVID-19 lockdown*. These methods were used to test a null hypothesis that there is no significant relationship between listed sociodemographic factors and the decision to change one's reproductive intentions after the COVID-19 lockdown. Fisher's exact

test was used when > 20% of expected cell counts was less than 5. Firth's logistic regression [21,22] was used to model the relationship between sociodemographic factors and changing of intentions. We chose to use Firth's logistic regression specifically due to the small sample size of participants who indicated that they did change their intentions, as Firth's logistic regression reduces bias or overestimation in the calculation of regression coefficients when working with smaller sample sizes or rare outcomes, especially in cases of separation or convergence. All statistical tests of hypotheses were done at a 5% significance level.

### Ethical considerations

The study was designed in accordance with the ethical principles outlined in the World Medical Association Declaration of Helsinki [23] and the Belmont Report [24]. These documents emphasize respect for individual's autonomy, justice, beneficence, and non-maleficence (do no harm) in the conduct of research with human participants. Ethical approval for the study was sought from the College of Humanities Ethical Review Committee, University of Ghana (Certified Protocol Number: ECH 108/ 20–21). This study was designed to ensure that the project does not expose participants to more than minimal risk (more than everyday risk). This study had measures in place to minimise the potential of harm to participants and was able to respond to any adverse consequences, i.e., emotional or psychological harm that may result from the sensitive questions (violence against women & recollection of possibly uncomfortable experiences during the pandemic) that were asked in this study (see below). These included provisions of complete privacy, use of hand-held computer tablets programmed to use unique identifiers only instead of names, use of highly experienced research assistants with extensive training on survey administration on sensitive topics such as VAW, interviewing of respondents in a sensitive and ethical manner, ready access to professional help (an identified social worker in selected sub-metropolitan areas and a clinical psychologist in both Greater Accra regional hospital and Komfo Anokye Teaching hospital), and assured (signed) confidentiality agreement with research assistants. Additionally, participants who demonstrated distress or reported being emotionally impacted by the research questions were withdrawn from the study by investigators and referred for professional psychological support. The contents (study objectives, potential risks, benefits, voluntary nature of participation, and confidentiality of information and study procedures) of the consent form were discussed with all participants in their preferred language (English, or local dialects). Parental/guardian consent was sought for adolescents below 18 years of age and the selected respondent was required to provide a separate assent for participation.

## Results

### Demographic and reproductive health characteristics of respondents

Of the 564 women interviewed for the overall study, 32 respondents were excluded from this analysis because they were above reproductive age. The remaining 532 were of reproductive age (generally defined as between the ages of 15–49, but the minimum age of respondents here was 19). The mean age (SD) of respondents was 34.95 (±7.73). Respondents were almost split evenly between Accra and Kumasi (47.9% and 52.1%), but the majority lived within their respective metropolis (78.0%). Most respondents (62.6%) belonged to the Akan ethnic group, and most were Christian (87.6%). The majority (53.2%) had received a primary education or less and were self-employed (55.3%). The majority (79.1%) were either in a relationship or had been within the preceding 12 months and nearly the same proportion (77.8%) had ever been married or cohabitated with a partner. The majority (82.5%) were pregnant at least once and most (60.5%) had between 1–3 children. Most (75.4%) desired more children than the number of living children they had, and most were not using contraception (60.9%) at the time of the survey. An equal number of respondents said that decisions to do with childbearing were made by themselves or by themselves and their partners (38.2% each). A plurality of respondents labeled their reproductive state prior to Covid lockdown as "undecided" (48.5%). Further details of demographic and reproductive health characteristics of respondents of reproductive age can be found in Table 1.

**Table 1. Demographic and reproductive health characteristics of respondents.**

| Characteristic | n (%) |
|---|---|
| **Age** | |
| 19-29 years old | 138 (25.9) |
| 30-39 years old | 225 (42.3) |
| 40-49 years old | 169 (31.8) |
| **City of residence** | |
| Accra | 255 (47.9) |
| Kumasi | 277 (52.1) |
| **Location of residence** | |
| Within the metropolis | 415 (78.0) |
| Outside of the metropolis | 117 (22.0) |
| **Ethnicity** | |
| Akan | 333 (62.6) |
| another ethnic group (Ga-Dangme, Ewe, Northern) | 199 (37.4) |
| **Ethnic group** | |
| Akan | 333 (62.6%) |
| Ga/Dangme | 69 (13.0%) |
| Ewe | 42 (7.9%) |
| **Religion** | |
| Christian | 466 (87.6) |
| Other (Muslim, traditional belief system, or no religion) | 66 (12.4) |
| **Religious group** | |
| Christian | 466 (87.6%) |
| Muslim | 61 (11.5%) |
| Other (traditional, Eastern, or not religious) | 5 (0.9%) |
| **Highest level of education completed** | |
| Primary education or less | 283 (53.2) |
| Secondary education | 148 (27.8) |
| Tertiary education | 101 (19.0) |
| **Employment status** | |
| No, | 59 (11.1) |
| Yes, self-employed | 294 (55.3) |
| Yes, employed by other | 179 (33.6) |
| **Income level in the past month** | |
| above median income of group (500 GHS) | 267 (50.2) |
| below median income of group (500 GHS) | 206 (38.7) |
| not applicable (not employed) | 59 (11.1) |
| **Have you been in a relationship within the last 12 months?** | |
| no | 111 (20.9) |
| yes | 421 (79.1) |
| **Have you ever been married or cohabitated for 6 months + ?** | |
| no | 118 (22.2) |
| yes | 414 (77.8) |
| **Ever pregnant** | |
| no | 93 (17.5) |
| yes | 439 (82.5) |

*(Continued)*

**Table 1.** (Continued)

| Characteristic | n (%) |
|---|---|
| **Number of living children** | |
| 0 | 124 (23.3) |
| 1-3 | 322 (60.5) |
| 3+ | 86 (16.2) |
| **Difference between reported number of children desired and number of living children** | |
| More children desired | 401 (75.4) |
| No more children desired | 131 (24.6) |
| **Current contraception use** | |
| no | 324 (60.9) |
| yes | 208 (39.1) |
| **Who makes decisions about how many children to have?** | |
| Myself only | 203 (38.2) |
| Myself and my partner | 203 (38.2) |
| My partner or someone else only | 33 (6.2) |
| Not applicable (never been pregnant) | 93 (17.5) |
| **Reproductive state prior to Covid lockdown** | |
| I was pregnant or expecting to become pregnant | 71 (13.3) |
| I was delaying or avoiding pregnancy | 203 (38.2) |
| Undecided | 258 (48.5) |
| **Total** | 532 (100) |

### Demographic and reproductive health characteristics of respondents by change in reproductive intentions after the COVID-19 lockdown

Of the 532 women interviewed who were of reproductive age, 25 (4.93%) reported that their reproductive intentions changed due to COVID-19 lockdowns. Women in Kumasi were more likely to report changing their intentions than women in Accra (7.9% v. 1.2%, p<0.001). Apart from city of residence, women living within their respective metropolis were more likely to report changing their intentions than women living outside of the metropolis (5.8% v. 0.9%, p<0.05). Those who had been married or who had lived with a partner in the past were more likely to report the change (6% v. 0%, p<0.01). A greater proportion of those whose childbearing decisions were made by other people changed their mind compared to those who made the decision entirely on their own (8.4% % 9.1% v. 2%, p<0.05). Finally, a greater proportion of those who were pregnant or expecting to become pregnant changed their mind compared to those who were delaying or avoiding pregnancy, or those who were undecided (16.9% v. 32% & 3.5%, p<0.001). There were no significant differences by age, ethnicity, religion, education, employment, income, pregnancy history, number of living children, or contraception use. Further details of demographic and reproductive health characteristics of those who reported changing their minds can be found in Table 2.

### Reported change in reproductive intentions following COVID-19 lockdown among respondents who reported changing their mind

Respondents who answered 'yes' to their reproductive plans changing since COVID-19 restrictions were imposed were asked what changed (Fig 1). Eight people responded 'undecided'. The next largest group reported that their change in intentions resulted in them getting pregnant–with 7 reporting doing so in accordance with their wishes and 4 reporting

**Table 2. Demographic and reproductive health characteristics of respondents by change in reproductive intentions after the COVID-19 lockdown.**

| | Did not change mind n (%)* | Changed mind n (%)* | P-Value |
|---|---|---|---|
| **Total** | 507 (95.3) | 25 (4.7) | |
| **Age** | | | 0.066 |
| 19-29 years old | 128 (92.8) | 10 (7.2) | |
| 30-39 years old | 213 (94.7) | 12 (5.3) | |
| 40-49 years old | 166 (98.2) | 3 (1.8) | |
| **City of residence** | | | <.001 |
| Accra | 252 (98.8) | 3 (1.2) | |
| Kumasi | 255 (92.1) | 22 (7.9) | |
| **Location of residence** | | | 0.026 |
| Within the metropolis | 391 (94.2) | 24 (5.8) | |
| Outside of the metropolis | 116 (99.1) | 1 (0.9) | |
| **Ethnicity** | | | 0.882 |
| Akan | 317 (95.2) | 16 (4.8) | |
| another ethnic group (Ga-Dangme, Ewe, Northern) | 190 (95.5) | 9 (4.5) | |
| **Religion** | | | 0.536 |
| Christian | 445 (95.5) | 21 (4.5) | |
| Other (Muslim, traditional belief system, or no religion) | 62 (93.9) | 4 (6.1) | |
| **Highest level of education completed** | | | 0.588 |
| Primary education or less | 272 (96.1) | 11 (3.9) | |
| Secondary education | 139 (93.9) | 9 (6.1) | |
| Tertiary education | 96 (95) | 5 (5) | |
| **Employment status** | | | 0.272 |
| No, | 54 (91.5) | 5 (8.5) | |
| Yes, self-employed | 280 (95.2) | 14 (4.8) | |
| Yes, employed by other | 173 (96.6) | 6 (3.4) | |
| **Income level in the past month**** | | | 0.431 |
| above median income of group (500 GHS) | 254 (95.1) | 13 (4.9) | |
| below median income of group (500 GHS) | 199 (96.6) | 7 (3.4) | |
| **Are you currently in a relationship or have you been in a relationship within the last 12 months?** | | | 0.540 |
| no | 107 (96.4) | 4 (3.6) | |
| yes | 400 (95) | 21 (5) | |
| **Have you ever been married or cohabitated for 6 months + ?** | | | 0.006 |
| no | 118 (100) | 0 (0) | |
| yes | 389 (94) | 25 (6) | |
| **Ever pregnant** | | | 0.101 |
| no | 92 (98.9) | 1 (1.1) | |
| yes | 415 (94.5) | 24 (5.5) | |
| **Number of living children** | | | 0.257 |
| 0 | 120 (96.8) | 4 (3.2) | |
| 1-3 | 303 (94.1) | 19 (5.9) | |
| 3+ | 84 (97.7) | 2 (2.3) | |
| **Difference between reported number of children desired and number of living children** | | | 0.305 |
| more children desired | 380 (94.8) | 21 (5.2) | |
| no more children desired | 127 (96.9) | 4 (3.1) | |

*(Continued)*

**Table 2.** (Continued)

| | Did not change mind n (%)* | Changed mind n (%)* | P-Value |
|---|---|---|---|
| **Current contraception use** | | | 0.456 |
| no | 307 (94.8) | 17 (5.2) | |
| yes | 200 (96.2) | 8 (3.8) | |
| **Who makes decisions about how many children to have?*** | | | 0.011 |
| Myself only | 199 (98) | 4 (2) | |
| Myself and my partner | 186 (91.6) | 17 (8.4) | |
| My partner or someone else only | 30 (90.9) | 3 (9.1) | |
| **Reproductive state prior to Covid lockdown** | | | <.001 |
| I was pregnant or expecting to become pregnant | 59 (83.1) | 12 (16.9) | |
| I was delaying or avoiding pregnancy | 199 (98) | 4 (2) | |
| Undecided | 249 (96.5) | 9 (3.5) | |

*%s across row

**Total values do not add up to the entire sample size because it excludes those who reported no current employment

***Total values do not add up to the entire sample size because it excludes those who had never been pregnant.

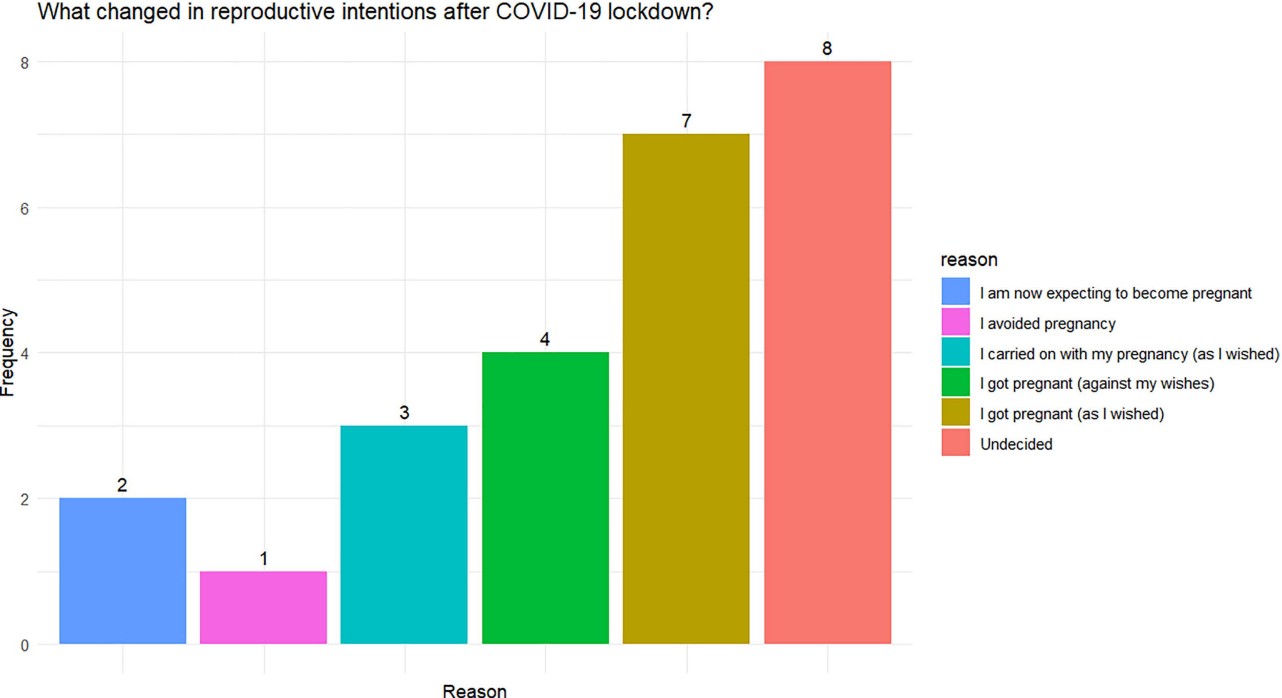

**Fig 1. Reported change in reproductive intentions following COVID-19 lockdown for respondents who reported changing their mind.**

doing so against their wishes. Three people reported changing their mind to carrying on with a pregnancy and 2 people reported changing their mind to conceiving soon. One person changed her intentions toward avoiding pregnancy.

**Effects of relevant participants demographic and reproductive health characteristics on changing reproductive intentions after COVID-19 lockdown.** Using Firth's logistic regression for rare events, a multivariate model was created

to evaluate the effects of various sociodemographic and reproductive health characteristic's effects on likelihood of changing of reproductive intentions after COVID-19 lockdown. Variables with perfect or near-perfect prediction of the outcome variable (location of residence, having ever been married or cohabitated, and having ever been pregnant), were omitted from the final model.

In the unadjusted model, women aged 19–29 had over three times the odds of changing their minds (OR=3.89; 95% CI 1.13–13.32) compared to the oldest respondents. Those living in Kumasi had over six times the odds of changing their minds (OR=6.35; 95% CI 2.03–19.86) compared to their counterparts in Accra. Women who reported others' involvement in decision-making regarding childbearing had over four times the odds of changing their plans when they made the decisions jointly with their partner (OR=4.16; 95% CI 1.45–11.95) and over five times the odds of changing their plans when it was their partner's or someone else's decision entirely (OR=5.09; 95% CI 1.20–21.64) compared to when they were solely responsible for the decision. Finally, those who were pregnant or expecting to become pregnant prior to lockdown had nearly ten times the odds of changing their plans (OR=9.31; 95% CI 3.04–28.42) compared to their counterparts who were delaying or avoiding pregnancy prior to lockdown.

However, once adjusted for other factors, only residence in Kumasi (AOR = 4.21, 95% CI 1.41–12.62) and being pregnant or expecting to become pregnant just before lockdown (AOR = 4.29, 95% CI 1.28–14.32) remained statistically significant predictors for changing of intentions (Table 3).

### Factors that informed reproductive intentions following the lockdown

All respondents were asked what the main factor that shaped their reproductive intentions was post-COVID-19 lockdown, whether or not they reported that they changed their minds. Out of 532 women who responded to questions regarding reproductive intentions, 44 (8.27%) cited reasons that had to do with COVID-19, including "fear & anxieties due to COVID," "financial crises due to COVID," "[not having] access to reproductive health services (contraception, ANC, abortion)," and "restrictions due to COVID-19." The remaining 488 respondents (91.73%) cited other reasons not related to COVID-19. Of the 532 women who responded to questions regarding reproductive intentions, 272 (51.12%) indicated that their plans were decided by themselves, their partner, a family member, or a combination. The other 260 (48.8%) indicated that their plans were not decided by anybody or that they were a result of a mistake.

Beyond the binary of changing one's intentions versus not changing one's intentions, reproductive intentions during this time were shaped by reasons both related and unrelated to COVID. As such, reasons for shaping of reproductive intentions, regardless of whether the respondent reported a change in intentions, can be broadly categorized as related to COVID-19 or unrelated to COVID-19. Further, in many cases, they were characterized by ambivalence, inconsistency, or a lack of intentionality. As such, shaping of reproductive intentions in this time can be conceptualized as a state arrived at by mistake or separate from specific decision making, or because of a decision made by oneself, one's partner, or one's family members. We note that these broad themes are not mutually exclusive.

For further descriptive detail on factors shaping reproductive intentions post-COVID-19 lockdown, see Table 4.

### Discussion

Overall, we found that few women reported changing their reproductive intentions due to the COVID-19 lockdown restrictions. This is in line with studies that assessed the effect of COVID-19 lockdown restrictions on reproductive intentions in other global regions [7,25,26].

The most significant factors affecting changing intentions among respondents were city and location of residence, history of marriage or cohabitation, pregnancy history, history of partner or other family member involvement in childbearing decisions, and reproductive state or intentions prior to COVID-19 outbreak. The effects of city and location of residence may be explained by differential acceptance of lockdown regulations [27], cultural differences by region, and by existing

**Table 3. Effects relevant participants demographic and reproductive health characteristics on changing reproductive intentions after COVID-19 lockdown.**

| Variables | Unadjusted analysis | | Adjusted analysis | |
|---|---|---|---|---|
| | OR | 95% CI | OR | 95% CI |
| **Age** | | | | |
| 19-29 years old | 3.89 | 1.13–13.32 | 3.35 | 0.83–13.47 |
| 30-39 years old | 2.79 | 0.84–9.27 | 3.03 | 0.97–9.46 |
| 40-49 years old (ref) | 1 | 1 | 1 | 1 |
| **City of residence** | | | | |
| Accra (ref) | 1 | 1 | 1 | 1 |
| Kumasi | 6.35 | 2.03–19.86 | 4.21 | 1.41–12.62 |
| **Religion** | | | | |
| Christian (ref) | 1 | 1 | 1 | 1 |
| other (Muslim, traditional belief system, or no religion) | 1.49 | 0.52–4.26 | 1.80 | 0.49–6.57 |
| **Highest level of education completed** | | | | |
| Primary education or less (ref) | 1 | 1 | 1 | 1 |
| Secondary education | 1.61 | 0.67–3.90 | 2.34 | 0.82–6.69 |
| Tertiary education | 1.35 | 0.48–3.83 | 1.51 | 0.43–5.36 |
| **Income level** | | | | |
| above median income (ref) | 1 | 1 | 1 | 1 |
| below median income | 1.41 | 0.57–3.51 | 1.18 | 0.46–3.00 |
| **Number of living children** | | | | |
| 0 (ref) | 1 | 1 | 1 | 1 |
| 1-3 | 1.72 | 0.60–4.90 | 2.32 | 0.50–10.84 |
| 3+ | 0.79 | 0.16–3.81 | 1.58 | 0.21–12.06 |
| **Difference between reported number of children desired and number of living children** | | | | |
| No more children desired (ref) | 1 | 1 | 1 | 1 |
| More children desired | 1.60 | 0.57–4.51 | 1.10 | 0.34–3.52 |
| **Who makes decisions regarding how many children to have?** | | | | |
| Myself only (ref) | 1 | 1 | 1 | 1 |
| Myself and my partner | 4.16 | 1.45–11.95 | 2.16 | 0.79–5.91 |
| My partner or someone else only | 5.09 | 1.20–21.64 | 2.18 | 0.47–10.00 |
| **Current contraception use** | | | | |
| No | 1.34 | 0.58–3.10 | 1.69 | 0.62–4.59 |
| Yes (ref) | 1 | 1 | 1 | 1 |
| **Reproductive state prior to COVID lockdown** | | | | |
| I was pregnant or expecting to become pregnant | 9.31 | 3.05–28.42 | 4.29 | 1.28–14.32 |
| I was delaying or avoiding pregnancy (ref) | 1 | 1 | 1 | 1 |
| Undecided | 1.69 | 0.54–5.26 | 1.08 | 0.33–3.58 |

differences in fertility rates and preferences in urban and peri-urban areas [28]. The effect of relationship status on changing of intentions may be related to income, security or stability afforded by being in a long-term relationship, and overall certainty of personal situation [29]. Isolation and financial challenges have been cited as reasons for postponement while cohabitation and subsequent closeness of relationships have been cited as reasons for acceleration [7]. The effect of reproductive state or intentions prior to outbreak on likelihood of changing intentions could reflect the ambivalence inherent in reproductive intentions in this population, suggesting that whatever the state was prior to lockdown determines actions following lockdown [12].

**Table 4. Factors that informed reproductive intentions following the lockdown.**

| | To do with COVID-19 (N=44) n (%) | Unrelated to COVID-19 (N=488) n (%) | Decided by nobody or by mistake (N=260) n (%) | Decided by self, partner, or family member (N=272) n (%) |
|---|---|---|---|---|
| **Age** | | | | |
| 19-29 years old | 9 (20.5) | 129 (26.4) | 60 (23.1) | 78 (28.7) |
| 30-39 years old | 18 (40.9) | 207 (42.4) | 110 (42.3) | 115 (42.3) |
| 40-49 years old | 17 (38.6) | 152 (31.1) | 90 (34.6) | 79 (29.0) |
| **City of residence** | | | | |
| Accra | 29 (65.9) | 226 (46.3) | 132 (50.8) | 123 (45.2) |
| Kumasi | 15 (34.1) | 262 (53.7) | 128 (49.2) | 149 (54.8) |
| **Location of residence** | | | | |
| Within the metropolis | 21 (47.7) | 394 (80.7) | 212 (81.5) | 203 (74.6) |
| Outside of the metropolis | 23 (52.3) | 94 (19.3) | 48 (18.5) | 69 (25.4) |
| **Ethnicity** | | | | |
| Akan | 33 (75.0) | 300 (61.5) | 164 (63.1) | 169 (62.1) |
| Another ethnic group (Ga-Dangme, Ewe, Northern) | 11 (25.0) | 188 (38.5) | 96 (36.9) | 103 (37.9) |
| **Religion** | | | | |
| Christian | 40 (90.9) | 426 (87.3) | 233 (89.6) | 233 (85.7) |
| Other (Muslim, traditional belief system, or no religion) | 4 (9.1) | 62 (12.7) | 27 (10.4) | 39 (14.3) |
| **Highest level of education completed** | | | | |
| Primary education or less | 26 (59.1) | 257 (52.7) | 146 (56.2) | 137 (50.4) |
| Secondary education | 14 (31.8) | 134 (27.5) | 72 (27.7) | 76 (27.9) |
| Tertiary education | 4 (9.1) | 97 (19.9) | 42 (16.2) | 59 (21.7) |
| **Employment status** | | | | |
| No, | 5 (11.4) | 54 (11.1) | 40 (15.4) | 19 (7.0) |
| Yes, self-employed | 35 (79.5) | 259 (53.1) | 141 (54.2) | 153 (56.3) |
| Yes, employed by other | 4 (9.1) | 175 (35.9) | 79 (30.4) | 100 (36.8) |
| **Income level in the past month** | | | | |
| Above median income of group (500 GHS) | 15 (34.1) | 252 (51.6) | 129 (49.6) | 138 (50.7) |
| Below median income of group (500 GHS) | 24 (54.5) | 182 (37.3) | 91 (35.0) | 115 (42.3) |
| Not applicable (not employed) | 5 (11.4) | 54 (11.1) | 40 (15.4) | 19 (7.0) |
| **Have you been in a relationship within the last 12 months?** | | | | |
| no | 4 (9.1) | 107 (21.9) | 49 (18.8) | 62 (22.8) |
| yes | 40 (90.9) | 381 (78.1) | 211 (81.2) | 210 (77.2) |
| **Have you ever been married or cohabitated for 6 months+?** | | | | |
| no | 2 (4.5) | 116 (23.8) | 62 (23.8) | 56 (20.6) |
| yes | 42 (95.5) | 372 (76.2) | 198 (76.2) | 216 (79.4) |
| **Ever pregnant** | | | | |
| no | 1 (2.3) | 92 (18.9) | 39 (15.0) | 54 (19.9) |
| yes | 43 (97.7) | 396 (81.1) | 221 (85.0) | 218 (80.1) |
| **Number of living children** | | | | |
| 0 | 2 (4.5) | 122 (25.0) | 55 (21.2) | 69 (25.4) |
| 1-3 | 38 (86.4) | 284 (58.2) | 165 (63.5) | 157 (57.7) |
| 3+ | 4 (9.1) | 82 (16.8) | 40 (15.4) | 46 (16.9) |

*(Continued)*

**Table 4.** (Continued)

| | To do with COVID-19 (N = 44) n (%) | Unrelated to COVID-19 (N = 488) n (%) | Decided by nobody or by mistake (N = 260) n (%) | Decided by self, partner, or family member (N = 272) n (%) |
|---|---|---|---|---|
| **Difference between reported number of children desired and number of living children** | | | | |
| More children desired | 41 (93.2) | 360 (73.8) | 199 (76.5) | 202 (74.3) |
| No more children desired | 3 (6.8) | 128 (26.2) | 61 (23.5) | 70 (25.7) |
| **Current contraception use** | | | | |
| No | 12 (27.3) | 312 (63.9) | 171 (65.8) | 153 (56.3) |
| Yes | 32 (72.7) | 176 (36.1) | 89 (34.2) | 119 (43.8) |
| **Who makes decisions regarding how many children to have?** | | | | |
| Decision made by partner or someone else | 3 (6.8) | 30 (6.1) | 21 (8.1) | 12 (4.4) |
| Decision made by self and partner | 14 (31.8) | 189 (38.7) | 94 (36.2) | 109 (40.1) |
| Decision made by self only | 26 (59.1) | 177 (36.3) | 106 (40.8) | 97 (35.7) |
| Missing | 1 (2.3) | 92 (18.9) | 39 (15.0) | 54 (19.9) |
| **Reproductive status prior to COVID lockdown** | | | | |
| I was delaying or avoiding pregnancy or not at risk of getting pregnant | 32 (72.7) | 171 (35.0) | 61 (23.5) | 142 (52.2) |
| I was pregnant or expecting to become pregnant | 4 (9.1) | 67 (13.7) | 43 (16.5) | 28 (10.3) |
| Undecided | 8 (18.2) | 250 (51.2) | 156 (60.0) | 102 (37.5) |

*Fear & anxieties due to COVID, financial crises due to COVID, lack of access to reproductive health services (contraception, ANC, abortion) and other restrictions due to COVID-19

**Fear and anxieties and financial crises unrelated to COVID-19, along with reports that nothing changed, expectations were met, or answers not applicable

While we did not find a significant relationship between income and changing of intentions, many studies have found that socioeconomic status is associated with fertility intentions. However, the majority of such studies have been done in high-income settings, and the relationship between socioeconomic status and fertility intentions is less well-understood in low- and middle-income settings [8]. One study that found decreased birth rates in the 8–10 months following the first state of emergency in Japan suggested that this is in line with already declining marriage and fertility rates in the country [30]. By contrast, a study of fertility intentions in Kenya found that those experiencing higher rates of food insecurity were more likely to accelerate childbearing, suggesting that childbearing may be seen as a stabilizing force [25]. In this vein, work studying fertility intentions in Ghana has found that job insecurity could encourage higher fertility as a way to insure the future [14]. This variation reflects the multitude of theories that underlie reproductive decision making. The New Home Economic theory suggests that childbearing is a rational economic choice [29]. By contrast, Friedman, et al. posed a theory that children may reduce uncertainty for women and enhance marital stability, so situations of financial insecurity might lead to higher rates of fertility, and that how a particular group responds to a disaster situation like the pandemic may depend on the demographic context prior to disaster [31]. The findings from this report suggest that interpersonal factors have a more significant relationship with reproductive intentions than economic ones among this population.

Studies that have attempted to further characterize lockdown-related changes in intention into pro- and anti-natal changes have found mixed effects. A study of Czech men and women one year after the initial COVID-19 lockdown found that 7% of survey respondents reduced their intended number of children due to the pandemic [32]. Meanwhile, a study conducted during the early lockdown in Nigeria found that approximately 20% of respondents considered the lockdown to have contributed to their latest pregnancy [33]. Our data, however, was largely characterized by ambivalence. While not

a majority, 48.5% of respondents, the largest single group, labeled their reproductive intentions prior to the lockdown as "undecided." Out of 25 people who reported changing their intentions, eight–again, the largest group–labeled the change as "undecided". Forty-four respondents cited COVID-19 as a major factor in their reproductive intentions, whether to do with fear and anxieties related to COVID, financial crises related to COVID, lack of access to reproductive health services, or other restrictions, even while only 25 people said that their minds changed because of COVID-19. Further, almost half of those interviewed–260 out of 532–said their intentions were decided by nobody or by mistake. It is possible that this ambivalence reflects a dissonance between desires and intention, with the lockdown's effects prompting a positive response to question of changed intentions that does not ultimately lead to a change in actions [34]. It could also suggest that there are other factors influencing childbearing that have yet to be characterized in the literature.

Overall, our findings indicate that few women explicitly changed their plans following the lockdown, demonstrating the necessity of ongoing robust sexual and reproductive health infrastructure as part of essential healthcare in crisis situations. While studies fortunately found that most healthcare systems recovered quickly from lockdown-related barriers to access, these findings also have implications for sexual and reproductive health messaging and care outside of the COVID-19 lockdown period.

The Demographic and Health Survey currently defines a "met need for family planning" as any method of contraceptive use in married or sexually active women between the ages of 15–49. The "unmet need for family planning" is defined as the proportion of these women who want to postpone their next birth by two or more years and are not using any method of contraception, or those who are having or have recently had a mistimed or unwanted birth. These definitions depend on clear-cut fertility intentions, following a two-year timeline and a binary consideration of timing and desire for fertility. However, our findings indicate that ambivalence and indecision are central to family planning among this population. This complexity at the individual level may mean that not all women can be classified accurately by the DHS questionnaire and that we may be overestimating the unmet need for family planning as defined by the questionnaire. Additionally, "contraception" includes methods ranging from sterilization to emergency contraception to unspecified traditional practices. A recently postpartum mother who relies on lactational amenorrhea at present but desires no future pregnancies may fall under the definition of someone with a "met need," although her choice of contraception may not serve her broader fertility goals, for example. This highlights the need for nuance in matching contraceptive choice to fertility goals.

## Strengths and limitations

This study has several strengths. A major strength is the large sample size. A threshold of 400 respondents is generally considered adequate for cross-sectional studies [35]. Out of 564 women surveyed, 532 were of reproductive age and responded to the series of questions regarding reproductive intentions. We were able to accommodate for a range of primary languages, including but not limited to English, Twi, and Ga. While participants were initially recruited by telephone, we were able to access those unreachable by phone using location information from the Ghana Statistical Service's COVID-19 business tracking survey. Another major strength is that this is one of the few studies of reproductive intentions to directly ask a question about changing of intentions due to a specific change. This may better capture how the shock of the pandemic restrictions specifically affected reproductive intentions. This approach may capture more granularity than demographic data as it narrows the question to the effects of a single environmental change.

However, this approach also comes with its limitations. Since the questionnaire was administered at one timepoint, there is possibility of recall bias as to what the reproductive status and intentions were prior to lockdown. Further, the short time interval between lockdown and data collection, at which time daily routines had not completely returned to a pre-pandemic "normal" means that we may have missed some changes in intentions relating to the lasting effects of COVID-19 and related restrictions. Our sample was restricted to women within the greater Accra and greater Kumasi areas, therefore the results may not be generalizable to other parts of Ghana, as rates of family planning usage, unintended pregnancies, and fertility intentions vary widely among the 16 regions.

## Future directions

When our study asked respondents to categorize the reasons for their fertility intentions following COVID-19 lockdowns, it did not allow much room for uncertainty or ambivalence. Given the literature suggesting that feelings of uncertainty or financial or marital insecurity impact childbearing intentions, future research should consider the effects of self-reported uncertainty levels on fertility intentions and changing intentions. Additionally, given the limited literature suggesting that Ghanaian women's fertility intentions are largely characterized by ambivalence, future work should incorporate more room for ambivalence in responses through qualitative approaches.

This work and future work regarding the complexity of reproductive intentions may inform how information is collected at the population level regarding fertility intentions and use of family planning methods. We hope that this information will in turn affect policy regarding the distribution of family planning services and approach to family planning counseling in the healthcare and public health settings. Additional information may encourage reproductive healthcare providers to prioritize patient-centered counseling that allows for a spectrum of reproductive goals with room for ambivalence and changing intentions over time. This information may also allow policymakers to gain a more accurate understanding of need in the realm of family planning, leading to more targeted interventions. Ultimately, this work may be used to inform policy and practice that is deeply rooted in patient autonomy and reproductive justice.

## Conclusion

This report adds to the existing literature exploring the impact of crisis situations like the COVID-19 lockdown on reproductive intentions using a diverse sample of Ghanaian women. We highlight a few demographic and reproductive health history factors that increased the likelihood of changing intentions following the lockdown. We also highlight the inherent ambivalence and inconsistencies in reproductive intentions among Ghanaian women, which is in line with the existing, albeit limited, literature investigating this population. These results suggest that the factors that contribute to reproductive intention is more nuanced and multifaceted. This may have implications for uptake of family planning and healthcare services and public health especially during crisis and emergencies. Therefore, reproductive healthcare services must take into consideration the nuances in reproductive intentions. Further, future study of reproductive intentions among Ghanaian women incorporates more room for uncertainty as we seek to address need for family planning and maternity care through a reproductive justice-minded approach. On the global level, the provision of healthcare services during emergencies should also take into consideration the uncertainties or women.

## Supporting information

**S1 File. Reported change in reproductive intentions following COVID-19 lockdown for respondents who reported changing their mind.**
(DOCX)

## Acknowledgments

Many thanks to the Ghana Statistical Services, particularly, Mrs. Joyce Date and Mr. Isaac Dadson, field research assistants, and other stakeholders without whom the study could not have been carried out.

## Author contributions

**Conceptualization:** Veena Bhagavathi, Deda Ogum, Kofi Agyabeng, Dorcas Coker-Appiah, Fidelia N. A. Ohemeng.

**Data curation:** Deda Ogum, Kofi Agyabeng, Fidelia N. A. Ohemeng.

**Formal analysis:** Veena Bhagavathi, Deda Ogum, Kofi Agyabeng, Fidelia N. A. Ohemeng.

**Funding acquisition:** Dorcas Coker-Appiah.

**Investigation:** Deda Ogum, Fidelia N. A. Ohemeng.

**Methodology:** Deda Ogum, Kofi Agyabeng, Fidelia N. A. Ohemeng.

**Project administration:** Deda Ogum.

**Resources:** Dorcas Coker-Appiah.

**Supervision:** Deda Ogum, Fidelia N. A. Ohemeng.

**Validation:** Veena Bhagavathi, Deda Ogum, Dorcas Coker-Appiah, Fidelia N. A. Ohemeng.

**Writing – original draft:** Veena Bhagavathi.

**Writing – review & editing:** Veena Bhagavathi, Deda Ogum, Kofi Agyabeng, Fidelia N. A. Ohemeng.

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
