## [Decision Letter · Decision Letter 0]

15 Feb 2025

PONE-D-24-25240Unchanged but undecided: Reproductive intentions among Ghanaian women following COVID-19 lockdowns in two large metropolises in GhanaPLOS ONE

Dear Dr. Ohemeng,

Thank you for submitting your manuscript to PLOS ONE. After careful consideration, we feel that it has merit but does not fully meet PLOS ONE’s publication criteria as it currently stands. Therefore, we invite you to submit a revised version of the manuscript that addresses the points raised during the review. Please submit your revised manuscript by Apr 01 2025 11:59PM. If you will need more time than this to complete your revisions, please reply to this message or contact the journal office at plosone@plos.org . Please include the following items when submitting your revised manuscript:

We look forward to receiving your revised manuscript.

Kind regards,

Obed Kwabena Offe Amponsah, PharmD, Ph.D.

Academic Editor

PLOS ONE

This study was funded by International Network to End Violence against Women and Girls (INEVAWG)

Many thanks to the International Network to End Violence Against Women and Girls (INEVAWG Network) for funding the study and the Gender Studies and Human Rights Documentation Centre for commissioning the study in Ghana. Thanks also to the Ghana Statistical Services, particularly, Mrs. Joyce Date and Mr. Isaac Dadson, our field research assistants, and other stakeholders without whom the study could not have been carried out.

This study was funded by International Network to End Violence against Women and Girls (INEVAWG)

7. We note that you have indicated that there are restrictions to data sharing for this study. For studies involving human research participant data or other sensitive data, we encourage authors to share de-identified or anonymized data. However, when data cannot be publicly shared for ethical reasons, we allow authors to make their data sets available upon request. For information on unacceptable data access restrictions, please see http://journals.plos.org/plosone/s/data-availability#loc-unacceptable-data-access-restrictions. 

8. When completing the data availability statement of the submission form, you indicated that you will make your data available on acceptance. We strongly recommend all authors decide on a data sharing plan before acceptance, as the process can be lengthy and hold up publication timelines. Please note that, though access restrictions are acceptable now, your entire data will need to be made freely accessible if your manuscript is accepted for publication. This policy applies to all data except where public deposition would breach compliance with the protocol approved by your research ethics board. If you are unable to adhere to our open data policy, please kindly revise your statement to explain your reasoning and we will seek the editor's input on an exemption. Please be assured that, once you have provided your new statement, the assessment of your exemption will not hold up the peer review process.

9. Please ensure that you refer to Figure 1 in your text as, if accepted, production will need this reference to link the reader to the figure.

10. Please remove all personal information, ensure that the data shared are in accordance with participant consent, and re-upload a fully anonymized data set. 

Reviewers' comments:

Reviewer's Responses to Questions

**Comments to the Author**

1. Is the manuscript technically sound, and do the data support the conclusions?

Reviewer #1: Yes

Reviewer #2: Yes

Reviewer #3: Yes

2. Has the statistical analysis been performed appropriately and rigorously? 

Reviewer #1: Yes

Reviewer #2: Yes

Reviewer #3: Yes

3. Have the authors made all data underlying the findings in their manuscript fully available?

Reviewer #1: Yes

Reviewer #2: Yes

Reviewer #3: Yes

4. Is the manuscript presented in an intelligible fashion and written in standard English?

Reviewer #1: Yes

Reviewer #2: Yes

Reviewer #3: Yes

5. Review Comments to the Author

Reviewer #1: The manuscript "Unchanged but undecided: Reproductive intentions among Ghanaian women following COVID-19 lockdowns in two large metropolises in Ghana" provides valuable insights into the impact of the COVID-19 pandemic on reproductive health intentions in Ghana. The study is highly relevant, addressing a critical topic with long-term global implications. By focusing on Ghanaian women in the metropolises of Accra and Kumasi, it adds a unique perspective to the understanding of how disasters such as the COVID-19 pandemic influence reproductive decision-making in low- and middle-income settings. The paper’s methodological rigor, including a substantial sample size (n=532) and the use of multivariate statistical techniques, strengthens its findings. The analysis provides a nuanced look at the factors influencing changes in reproductive intentions, particularly the role of pre-existing pregnancy intentions and socio-demographic factors such as residence and marital status. Furthermore, the novel finding of widespread ambivalence in reproductive intentions, with many women remaining undecided both before and after the lockdown, contributes new knowledge to the field. The study’s results are particularly valuable for informing reproductive health policy and interventions in crisis settings. Overall, this manuscript represents a solid contribution to reproductive health research, deserving acceptance with minor revisions related to language and further elaboration of the discussion.

Reviewer #2: Reviewer’s comments

Abstract

1. The abstract mentions the potential influence of pre-existing social context but does not state the specific factors (e.g., economic status, education, cultural norms) may have influenced reproductive intentions.

2. The abstract does not specify the research method and research design used in the study e.g. purely quantitative or mixed method. I suggest the authors should clarify this.

3. The abstract does not state the specific statistical techniques used e.g. the multivariate model, logistic regression, multiple regression etc. which would help clarify the analysis.

4. The abstract mentions the need for future work to accommodate more ambiguity in characterizing reproductive intentions, it does not elaborate on the broader implications of these findings for policy or public health interventions.

Introduction

1. The first sentence is incomplete ("met with a swift pandemic shook..."). This sentence needs to be revised for clarity and to ensure it flows logically.

2. The phrase "fertility intentions" is mentioned, it might be useful to explain it briefly earlier, particularly for readers who may not be familiar with the term.

3. ……………” Some people were overworked, taking on additional childcare on top of remote jobs” What does this sentence mean? I suggest that the authors revise the sentence to make it clearer for readers to understand.

4. While some studies are referenced, there is no explicit mention of the research gap this study addresses. The introduction could more clearly explain what is lacking in current literature and why this study is needed.

5. This study sought to evaluate if and why Ghanaian women changed their reproductive

6. intentions because of COVID-19 lockdown measures. This sentence is hanging. Please, check the sentence.

Research methods

Research design

1. “This was an analytical cross-sectional study that used a convergent mixed methods design to primarily assess the economic and social impact of COVID-19 response measures on women and girls in Accra and Kumasi”…... Do you mean a "convergent mixed methods design"? If so, how is this design applied within a single study like this, particularly since the authors did not specify it in the abstract? This design is typically applicable in mixed-methods studies. I suggest the authors clarify this aspect to strengthen their explanation.

2. How was this design applied in the study? The design appears inappropriate and, also seems isolated and not effectively integrated into the study. I suggest the authors thoroughly revise this section.

Study area

1. The choice of Accra and Kumasi is logical given their high case counts and economic significance, the justification would be stronger if the authors explicitly addressed issues of representation, regional comparisons, and broader applicability, it will strengthen the study setting.

Recruitment of participants

1. Combination of sampling techniques. The random, systematic, and purposeful sampling techniques were used, the details of how these methods were combined are unclear

2. Is this study quantitative or mixed-method? It seems like the study does not follow either approach.

3. Additionally, the study does not provide details on how the different sampling methods (random, systematic, and purposeful) were applied in the data collection process

4. “The exclusion criteria were non-residence and not being economically engaged in the study area for at least 6 months prior to initiation of COVID-19 lockdown and 6 months following the lockdown”. How were the participants excluded, and why were these participants excluded?"

5. The participants recruitment lacks clarity and justification. I suggest the authors should revise this section.

Study tool

1. ‘Information was collected with ODK software using a 5-section structured questionnaire” What do you mean by ODK software and 5-section structured questionnaire?

2. How was the tool used to gather the data?

3. Include how piloting improves the data collection process.

Data analysis

1. There is no explanation of why Chi-squared and Fisher’s exact tests were chosen for hypothesis testing or why Firth’s logistic regression was used instead of other statistical techniques

2. Chi-squared and Fisher's exact tests examine associations between categorical variables, but they do not account for confounders or control for multiple variables simultaneously. How did you account for or address confounding variables?

3. There is no clear explanation of how the hypothesis testing was conducted, including the null hypothesis, alternative hypothesis, and criteria for significance

Results

1. The presentation of the results is generally good; however, the authors should revise this section to avoid redundancy.

Discussion.

1. The discussion is good but lacks implications for policy and practice. I suggest the authors include these aspects in the discussion.

Conclusion

1. The conclusion does not mention how the findings can be applied to inform policy or practice, especially in areas such as family planning, healthcare services, or public health interventions

2. There is no mention of the limitations of the study, which is a critical component of the conclusion.

3. The conclusion briefly touches on the need for future studies, but it lacks specific recommendations for further research in this area

4. The conclusion references the COVID-19 lockdown, it doesn't connect the study's findings to broader trends or global implications.

Reviewer #3: 1. Though the study presents the Ghana case, it must give an insight into the global context on women reproductive intentions because of the Covid-19 lock down measures.

2. Add 'sexual and reproductive healthcare' to the keywords

3. Under the methods, describe the rigorous methods used in conducting the telephone interviews.

4. There is the need to introduce some thick quotes from the qualitative data to support the quantitative analysis.

5. Break down the percentages for the 'other' in ethnicity and religion. Discuss scholarly the socio-demographics and how they could have potentially affected the results garnered.

6. The concluding section is the weakest section. It has not presented any solid conclusions based on the study's results.

7. Write in past tense as you are reporting on an already conducted study.

6. PLOS authors have the option to publish the peer review history of their article (what does this mean? ). If published, this will include your full peer review and any attached files.

**Do you want your identity to be public for this peer review?** For information about this choice, including consent withdrawal, please see our Privacy Policy .

Reviewer #1: **Yes: ** sara Akram

Reviewer #2: No

Reviewer #3: **Yes: ** Dickson Adom

---

## [Author Response · Author response to Decision Letter 1]

11 Apr 2025

The responses to the reviewers comments have been uploaded as a supporting document.

---

## [Decision Letter · Decision Letter 1]

12 May 2025

PONE-D-24-25240R1Unchanged but undecided: Reproductive intentions among Ghanaian women following COVID-19 lockdowns in two large metropolises in GhanaPLOS ONE

Dear Dr. Ohemeng,

Thank you for submitting your manuscript to PLOS ONE. After careful consideration, we feel that it has merit but does not fully meet PLOS ONE’s publication criteria as it currently stands. Therefore, we invite you to submit a revised version of the manuscript that addresses the points raised during the review process.

**ACADEMIC EDITOR: **Once again, thank you for submitting the revised manuscript based on the reviewer comments, all of who agree great progress has been made. This round of reviews requires a minor revision mainly to the methods section as detailed in the reviewer comments below. Kindly address them completely as soon as possible to facilitate potential publication of this body of work.

We look forward to receiving your revised manuscript.

Kind regards,

Obed Kwabena Offe Amponsah, PharmD, Ph.D.

Academic Editor

PLOS ONE

Journal Requirements:

Reviewers' comments:

Reviewer's Responses to Questions

**Comments to the Author**

1. If the authors have adequately addressed your comments raised in a previous round of review and you feel that this manuscript is now acceptable for publication, you may indicate that here to bypass the “Comments to the Author” section, enter your conflict of interest statement in the “Confidential to Editor” section, and submit your "Accept" recommendation.

Reviewer #1: All comments have been addressed

Reviewer #2: All comments have been addressed

Reviewer #3: (No Response)

2. Is the manuscript technically sound, and do the data support the conclusions?

Reviewer #1: Yes

Reviewer #2: Yes

Reviewer #3: Yes

3. Has the statistical analysis been performed appropriately and rigorously? 

Reviewer #1: Yes

Reviewer #2: Yes

Reviewer #3: Yes

4. Have the authors made all data underlying the findings in their manuscript fully available?

Reviewer #1: Yes

Reviewer #2: (No Response)

Reviewer #3: Yes

5. Is the manuscript presented in an intelligible fashion and written in standard English?

Reviewer #1: Yes

Reviewer #2: Yes

Reviewer #3: Yes

6. Review Comments to the Author

Reviewer #1: good luck for this research publication. I have reviewed the revised version of the manuscript titled "Unchanged but undecided: Reproductive intentions among Ghanaian women following COVID-19 lockdowns in two large metropolises in Ghana" (Manuscript Number: PONE-D-24-25240R1).

The authors have adequately addressed the concerns and suggestions raised in the initial review. The revisions have strengthened the clarity and quality of the manuscript, and I believe it is now suitable for publication.

Reviewer #2: The comments have been successfully addressed; however, there are a few remaining points for the authors to address.

1. Abstract.

The abstract is clear and comprehensive; however, the phrase 'quantitative research design' in line 28 is too broad. I suggest that the authors specify which type of quantitative research design was employed, such as descriptive quantitative research design, correlational quantitative research design, etc.

2. Results.

“The prepositional phrase “of the”......is used frequently throughout the work (e.g., in line 28, in line 254, in line 328, etc.). I suggest that the authors vary the usage of this prepositional phrase to avoid monotony.

3. The limitations

The authors claimed in their response that a subheading on the limitations has been included. However, there is no subheading on limitations addressing the comment.

Reviewer #3: Thanks for revising the manuscript based on my review report. The revised version is much stronger and scholarly. However, since you clarified that the manuscript was from the quantitative aspect of a larger convergent parallel mixed methods study, I suggest you rewrite the methods section as such. Just state the broad research approach adopted for study so you don't unnecessarily confuse readers. Also, give a more nuanced explanation of your choices of the methods and their appropriateness for the study.

7. PLOS authors have the option to publish the peer review history of their article (what does this mean? ). If published, this will include your full peer review and any attached files.

**Do you want your identity to be public for this peer review?** For information about this choice, including consent withdrawal, please see our Privacy Policy .

Reviewer #1: **Yes: ** sara Akram

Reviewer #2: **Yes: ** Dr Awinaba Amoah Adongo

Reviewer #3: **Yes: ** Dickson Adom

---

## [Author Response · Author response to Decision Letter 2]

12 Jun 2025

Dear Editor,

RESPONSE TO REVIEWER COMMENTS

We wish to thank you and all reviewers of our manuscript. We strongly believe that your suggestions for revision have greatly improved the current draft of the manuscript. Below are a point-by-point responses to all comments given:

Reviewer #1:

Good luck for this research publication. I have reviewed the revised version of the manuscript titled "Unchanged but undecided: Reproductive intentions among Ghanaian women following COVID-19 lockdowns in two large metropolises in Ghana" (Manuscript Number: PONE-D-24-25240R1).

The authors have adequately addressed the concerns and suggestions raised in the initial review. The revisions have strengthened the clarity and quality of the manuscript, and I believe it is now suitable for publication.

Response: We wish to thank the reviewer for their scrutiny of the manuscript.

Reviewer #2: The comments have been successfully addressed; however, there are a few remaining points for the authors to address.

1. Abstract.

The abstract is clear and comprehensive; however, the phrase 'quantitative research design' in line 28 is too broad. I suggest that the authors specify which type of quantitative research design was employed, such as descriptive quantitative research design, correlational quantitative research design, etc.

Response: Thank you for pointing this out. The text has been updated to specify the research design as “correlational quantitative research design”. Please see line 28.

2. Results.

“The prepositional phrase “of the”......is used frequently throughout the work (e.g., in line 28, in line 254, in line 328, etc.). I suggest that the authors vary the usage of this prepositional phrase to avoid monotony.

Response: We thank the reviewer for the feedback. We have reviewed the entire manuscript and varied text previously containing the phrase to eliminate the phrase if not absolutely necessary.

3. The limitations

The authors claimed in their response that a subheading on the limitations has been included. However, there is no subheading on limitations addressing the comment.

Response: Thank you for your comment. Please see line 430 of the manuscript for the Strengths & Limitations subheading as previously submitted.

Reviewer #3:

Thanks for revising the manuscript based on my review report. The revised version is much stronger and scholarly. However, since you clarified that the manuscript was from the quantitative aspect of a larger convergent parallel mixed methods study, I suggest you rewrite the methods section as such. Just state the broad research approach adopted for the study so you don't unnecessarily confuse readers. Also, give a more nuanced explanation of your choices of methods and their appropriateness for the study.

Response: We wish to thank the reviewer for pointing out the possible confusion that readers may have with situating methods of the current study within that of the larger study from which the data was obtained. We have updated sections under study design and study population to eliminate possibly confusing aspects pertaining to the larger study and facilitate reader understanding of the current study. Please see lines 114-119; 140-148

---

## [Decision Letter · Decision Letter 2]

30 Jun 2025

Unchanged but undecided: Reproductive intentions among Ghanaian women following COVID-19 lockdowns in two large metropolises in Ghana

PONE-D-24-25240R2

Dear Dr. Ohemeng,

We’re pleased to inform you that your manuscript has been judged scientifically suitable for publication and will be formally accepted for publication once it meets all outstanding technical requirements.

Kind regards,

Obed Kwabena Offe Amponsah, PharmD, Ph.D.

Academic Editor

PLOS ONE

Additional Editor Comments (optional):

Reviewers' comments:

Reviewer's Responses to Questions

**Comments to the Author**

1. If the authors have adequately addressed your comments raised in a previous round of review and you feel that this manuscript is now acceptable for publication, you may indicate that here to bypass the “Comments to the Author” section, enter your conflict of interest statement in the “Confidential to Editor” section, and submit your "Accept" recommendation.

Reviewer #2: All comments have been addressed

Reviewer #3: All comments have been addressed

2. Is the manuscript technically sound, and do the data support the conclusions?

Reviewer #2: Yes

Reviewer #3: Yes

3. Has the statistical analysis been performed appropriately and rigorously? 

Reviewer #2: Yes

Reviewer #3: Yes

4. Have the authors made all data underlying the findings in their manuscript fully available?

Reviewer #2: Yes

Reviewer #3: Yes

5. Is the manuscript presented in an intelligible fashion and written in standard English?

Reviewer #2: Yes

Reviewer #3: Yes

6. Review Comments to the Author

Reviewer #2: (No Response)

Reviewer #3: Thanks for the clarity in the methods section in this revised manuscript following my earlier query. Also, the study's limitations have been well articulated. All the best

7. PLOS authors have the option to publish the peer review history of their article (what does this mean? ). If published, this will include your full peer review and any attached files.

**Do you want your identity to be public for this peer review?** For information about this choice, including consent withdrawal, please see our Privacy Policy .

Reviewer #2: **Yes: ** Dr Awinaba Amoah Adongo

Reviewer #3: **Yes: ** Dickson Adom
